# On-the-fly Point Feature Representation for Point Clouds Analysis

## ABSTRACT

Point cloud analysis is challenging due to its unique characteristics of unorderness, sparsity and irregularity. Prior works attempt to capture local relationships by convolution operations or attention mechanisms, exploiting geometric information from coordinates implicitly. These methods, however, are insufficient to describe the explicit local geometry, *e.g.*, curvature and orientation. In this paper, we propose *On-the-fly Point Feature Representation (OPFR)*, which captures abundant geometric information explicitly through *Curve Feature Generator* module. This is inspired by Point Feature Histogram (PFH) from computer vision community. However, the utilization of vanilla PFH encounters great difficulties when applied to large datasets and dense point clouds, as it demands considerable time for feature generation. In contrast, we introduce the *Local Reference Constructor* module, which approximates the local coordinate systems based on triangle sets. Owing to this, our OPFR only requires extra *1.56ms* for inference (**65×** faster than vanilla PFH) and *0.012M* more parameters, and it can serve as a versatile plug-and-play module for various backbones, particularly MLP-based and Transformer-based backbones examined in this study. Additionally, we introduce the novel *Hierarchical Sampling* module aimed at enhancing the quality of triangle sets, thereby ensuring robustness of the obtained geometric features. Our proposed method improves overall accuracy (OA) on ModelNet40 from 90.7% to **94.5**% (+3.8%) for classification, and OA on S3DIS Area-5 from 86.4% to **90.0**% (+3.6%) for semantic segmentation, respectively, building upon PointNet++ backbone. When integrated with Point Transformer backbone, we achieve state-of-the-art results on both tasks: **94.8**% OA on ModelNet40 and **91.7**% OA on S3DIS Area-5.

## CCS CONCEPTS

• **Computing methodologies** → **Scene understanding**; **Computer vision representations**.

## KEYWORDS

Scene understanding, point clouds representation, local geometry, classification, semantic segmentation.

## 1 INTRODUCTION

Point cloud analysis on robotics and automation application [15, 28, 51, 53, 62] has garnered substantial attention in recent years,

Permission to make digital or hard copies of all or part of this work for personal or classroom use is granted without fee provided that copies are not made or distributed for profit or commercial advantage and that copies bear this notice and the full citation on the first page. Copyrights for components of this work owned by others than the author(s) must be honored. Abstracting with credit is permitted. To copy otherwise, or republish, to post on servers or to redistribute to lists, requires prior specific permission and/or a fee. Request permissions from permissions@acm.org.
*ACM MM, 2024, Melbourne, Australia*
© 2024 Copyright held by the owner/author(s). Publication rights licensed to ACM.
ACM ISBN 978-x-xxxx-xxxx-x/YY/MM
https://doi.org/10.1145/nnnnnnn.nnnnnnn

driven by advancements in sensor technologies like LiDAR and photogrammetry. This growing interest attributes to two key advantages: 1) It can accurately represent complex objects with numbers of points. 2) It can be quickly created by using 3D scanning devices. Compared to 2D image data, point clouds provide a more powerful 3D sparse representation containing abundant geometry and layout information of the environment.

Deep learning technology [10, 18] has achieved significant improvements in various image processing tasks. However, the typical deep learning technology requires highly regular input data formats. The unordered and irregular point clouds bring great challenges to apply the image processing techniques directly. PointNet [30], the pioneering work of network architecture that directly works with point clouds, overcomes the challenges of the unordered and irregular inputs. It uses point-wise shared-MLP followed by a pooling operation to extract global features from point clouds, but global pooling operation leads to the loss of valuable local information. PointNet++ [31] further proposes set abstraction (SA) to process local regions hierarchically. This step aggregates features from neighboring points, thereby capturing local information. However, it still learns from individual points without incorporating local relationships [21]. This could hinder the model from leveraging inherent point clouds geometric structures.

Local geometric structures are vital for understanding point clouds. In an effort to capture this information, some prior works attempt to learn local relationships from convolutions [14, 20, 54], attentions [8, 60, 65], or graphs [50, 59, 63]. However, these methods require huge amount of labelled data to learn local geometry implicitly [33], while getting large amount of labelled 3D annotations is difficult. Recently, RepSurf [33] has emerged as a novel approach that explicitly learns geometric information based on umbrella surface [6], which is a triangle set[1] with connected triangles formed by $k$ nearest neighbors ($k$-NNs). While triangle sets are effective in capturing location and orientation information, they often fall short in incorporating curvature knowledge, which is essential for accurate point cloud recognition [2, 42]. Moreover, as depicted in the supplementary material, in certain $k$-NNs, the $k$ points may come from different surfaces of the object. These "noisy points" can lead to the distortion of $k$-NN triangle sets, significantly impacting the quality of the obtained geometric features [33].

To integrate curvature information explicitly, we draw inspiration from Point Feature Histogram (PFH) [37], a notable handcrafted feature descriptor for capturing regional curvature knowledge. PFH exploits the histogram of curvature angles within local neighborhoods to characterize individual points. As shown in Fig. 1, these angles are calculated between the normal vector and the local coordinate system, which demand substantial computing resources. Nonetheless, many point cloud datasets [7, 48] lack normal vectors

---

[1]In this paper, we refer triangle set to a collection of connected or disconnected triangles. For those connected ones, they form a surface.

and necessitate additional normal estimation [12, 13, 38]. Normal estimation poses significant computational challenges, particularly for dense point clouds, while its accuracy degenerates considerably for sparse point clouds. These limitations can potentially lead to the breakdown of vanilla PFH approach, further underscoring the challenges of its direct integration with deep learning models.

In view of PFH's potentials and limitations, we explore curvature information and propose **On-the-fly Point Feature Representation (OPFR)**, which includes *Local Reference Constructor* module and *Curve Feature Generator* module. This provides an efficient way to leverage explicit curvature knowledge without the prerequisite of normal estimation, which inherently relies on the quality of triangle sets. Additionally, we propose the novel *Hierarchical Sampling* module to mitigate the distortion of triangle sets that occurs in the naive $k$-NN approach. Our sampling method demonstrates the robustness against noisy points by employing a hierarchical sampling strategy and a farthest point sampling strategy. As a result, it can significantly improve the obtained geometric features. These innovations confer the following properties:

- **Curvature Awareness.** The usage of curvature information remains underexplored by prior works. Our proposed OPFR obtains the capability to explicitly capture not only location and orientation knowledge, but also curvature geometry via *Curve Feature Generator* module.

- **Computational Efficiency.** Vanilla PFH is computationally expensive due to normal estimation. Our proposed OPFR introduces *Local Reference Constructor* module, which approximates the local coordinate systems based on triangle sets to overcome the computational bottlenecks.

- **Robustness.** Naive $k$-NN sampling causes distortion of triangle sets, which compromises the obtained geometric features. In contrast, our proposed OPFR presents *Hierarchical Sampling* module to enhance the quality of triangle sets, ensuring robust geometric features for noisy points.

Moreover, our OPFR is backbone-agnostic, making it compatible with different 3D point clouds analysis architectures. We demonstrate its model-agnostic nature by adapting two representative backbones: PointNet++ [31] and Point Transformer [65]. It serves as an efficient plug-and-play module, and achieves substantial performance improvements. Empirical results prove its compatibility with different backbones. When incorporating with Point Transformer backbone, our OPFR achieves state-of-the-art performance for both point cloud classification and semantic segmentation tasks.

## 2 RELATED WORK

### 2.1 Deep Learning on Point Clouds

Many prior works [4, 22, 27, 30, 31, 61, 64] learn from raw point clouds via careful network designs. PointNet [30] pioneers this trail by handling coordinates of each point with shared-MLP and consolidating the final representation with a global pooling operation. However, it is susceptible to a deficiency in preserving local structures due to the use of global pooling operation. PointNet++ [31] is an extension of the original PointNet architecture, which applies PointNet to multiple subsets of point clouds. It further leverages a hierarchical feature learning paradigm to capture the local structures. However, PointNet++ still processes points individually in

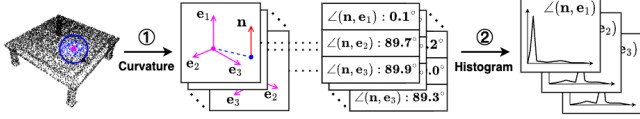

**Figure 1: The workflow of Point Feature Histogram, which can be decomposed into two steps. Firstly, for one interested point (pink), each neighboring point (blue) pair is described via angles. Secondly, for each angle, its distribution over all point pairs is summarized using histograms. Here, $(\mathbf{e}_1, \mathbf{e}_2, \mathbf{e}_3)$ is the local coordinate system and $\mathbf{n}$ is the normal vector.**

each local region, neglecting explicit consideration of relationships between centroids and their neighbors.

As PointNet++ establishes the hierarchical point clouds analysis framework, the focus of many works has shifted towards the development of local feature extractors, including convolution-based [14, 17, 20, 25, 54], attention-based [8, 60, 65], and graph-based [46, 50, 58, 59, 63] approaches. PointCNN [20] learns a $\chi$-transformation from input point clouds, which attempts to re-organize inputs into canonical order. Subsequently, it utilizes vanilla convolution operations to extract local features. Point Transformer [65] replaces conventional shared-MLP modules with Transformer [49] blocks, serving as feature extractors within localized patch processing. DGCNN [50] utilizes dynamic graph structures to enhance feature learning and capture relationships between points. However, these works rely heavily on the learnability of feature extractors, potentially missing inherent local shape information. More recently, RepSurf [33] leverages triangle sets with connected triangles formed by $k$ nearest neighbors ($k$-NNs), to learn location and orientation-aware representations from geometric features explicitly. Although location and orientation features are explicitly injected into network architecture in RepSurf, the usage of curvature information still remains underexplored. Moreover, RepSurf relies on naive $k$-NNs to produce triangle sets and obtain geometric features, which are vulnerable to noisy points [33].

### 2.2 Hand-crafted Designs on Point Clouds

Many works in 3D computer vision attempt to build sophisticated feature descriptors [35, 37, 39], which help to understand point clouds through hand-crafted features. Point Feature Histogram (PFH) [37], one of the feature descriptors, is commonly used in computer vision tasks like object recognition, registration and model retrieval [11, 19, 36]. It develops point cloud representations by summarizing the distribution of certain geometric attributes within a local neighborhood around each point.

We depict the workflow of PFH derivation for one of our interested point in Fig. 1. The whole process can be decomposed into two steps. Firstly, for each point pair within $k$ nearest neighbors ($k$-NNs) of interested point, curvature features are characterized using angles calculated from normal vectors and relative positions. Secondly, for each angle, we achieve its histogram within $k$-NNs. The histograms of different angles are concatenated together, yielding the final PFH representation. Unfortunately, many point cloud datasets [7, 48] collected in real-world scenarios lack normal vectors, and

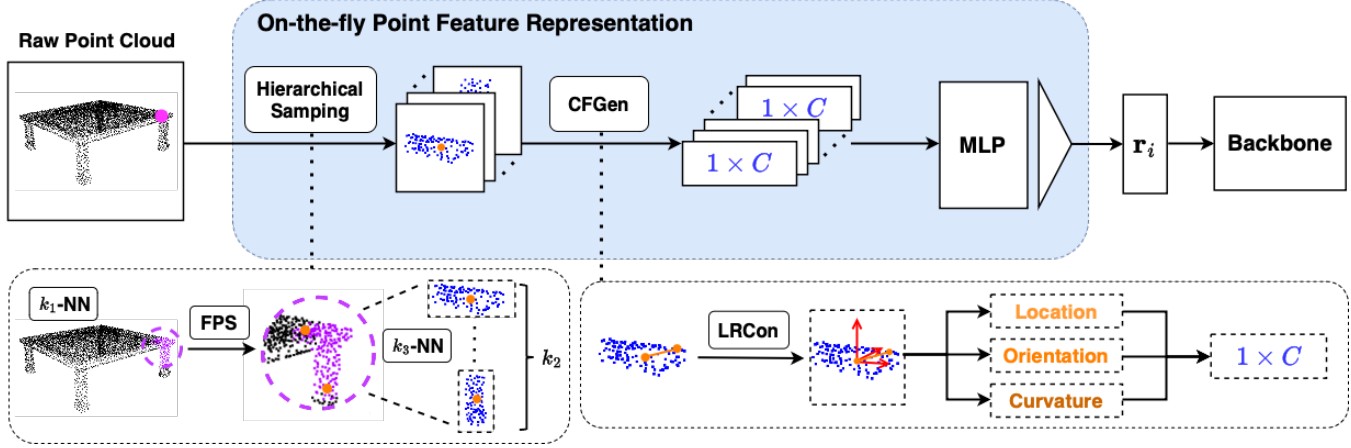

**Figure 2: Illustration of On-the-fly Point Feature Representation (OPFR) learning paradigm. The generation of geometric features consists of three key modules: *Hierarchical Sampling, Local Reference Constructor (LRCon)* and *Curve Feature Generator (CFGen)*. These geometric features are further fed into shared-MLP followed by pooling operation, constituting the final OPFR.**

estimating normal vectors [13, 52] for sparse point clouds often leads to significant deviations from ground truth. Nonetheless, PFH calculation involves establishing local coordinate systems and constructing curvature features, which is computationally expensive. As a result, the practical application of vanilla PFH is limited.

## 3 METHODOLOGY

The pipeline of **On-the-fly Point Feature Representation (OPFR)** is depicted in Fig. 2, and we illustrate the OPFR generating process for the right corner of table (highlighted in pink), which is one of our interested points. Firstly, we propose *Hierarchical Sampling* module, which takes each point in the point cloud as input and outputs several clusters (highlighted in blue) and corresponding centroids (highlighted in orange). This hierarchical sampling strategy improves the quality of triangle set for each point, thereby facilitating the development of subsequent geometric features. Then, for each point pair within the clusters, we design *Curve Feature Generator* module to generate geometric features, including location, orientation, and curvature. The inclusion of explicit curvature information allows us to more effectively capture the local geometry surrounding these point pairs. To enhance the efficiency and enable on-the-fly processing, we present *Local Reference Constructor* module. It approximates a local coordinate system (highlighted in red) for each point pair using adjacent points from triangle sets. Lastly, these obtained geometric features are further fed into shared-MLP followed by a pooling operation, constituting our final OPFR representation $\mathbf{r}_i$. The resultant OPFR representation $\mathbf{r}_i$ along with coordinate $\mathbf{x}_i$ can be directed into various point clouds analysis backbones, *e.g.*, PointNet++ [31] and Point Transformer [65], for end-to-end training.

### 3.1 Hierarchical Sampling

As mentioned earlier, the connected triangle sets produced by naive $k$ nearest neighbors ($k$-NNs) are susceptible to noisy points [33], leading to significant distortion. Given that our *Local Reference*

**Algorithm 1** Pseudo-code of **Hierarchical Sampling**

```python
def Hierarchical_Sampling(pc, k1, k2, k3):
    # k1: number of nearest neighbors for each point
    # k2: number of centroids within k1 nearest neighbors
    # k3: number of neighbors for each selected centroid
    # pc: input point clouds        # [B, N, 3]
    nearest_neighbors = kNN(inputs=pc, k=k1)
                                    # [B, N, k1, 3]
    selected_centroids = FPS(inputs=nearest_neighbors, k=k2)
                                    # [B, N, k2, 3]
    surface_points = kNN(inputs=selected_centroids, k=k3)
                                    # [B, N, k2, k3, 3]
    return surface_points
```

*Constructor* module inherently relies on triangle sets to approximate local reference frames, we propose the novel *Hierarchical Sampling* module to alleviate this distortion issue. For each individual point, our *Hierarchical Sampling* module generates several clusters, forming a triangle set. Specifically, we firstly conduct $k$-NN algorithm to select the $k_1$ nearby points (highlighted in purple). Secondly, we utilize farthest point sampling [5] algorithm to identify $k_2$ surface centroids (highlighted in orange) from these $k_1$ nearby points. Lastly, for each centroid, we retrieve its $k_3$ nearest neighbors (highlighted in blue). The selected $k_3$ neighbors are used to further develop geometric features. The detailed implementation is presented in Algorithm 1.

As illustrated in Fig. 2, *Hierarchical Sampling* module is designed to decouple the right corner of the table into $k_2$ distinct clusters (*e.g.*, table top and table leg). These $k_2$ clusters exhibit simpler geometric structures, allowing resultant triangle sets to better approximate the original local surface. Therefore, compared to the naive $k$-NN approach, our hierarchical sampling scheme effectively releases the distortion issue of triangle sets, and ensures the robustness against original noisy points. As a result, it greatly enhances the development of subsequent geometric features. We provide additional visualization examples comparing triangle sets generated by *Hierarchical Sampling* and those produced by $k$-NN sampling in the supplementary material.

## 3.2 Local Reference Constructor

A local reference frame is a local system of Cartesian coordinates at each point [26], which provides a reference for understanding local structures. Denote a point set as $\mathbf{X} = \{\mathbf{x}_1, \mathbf{x}_2, \cdots, \mathbf{x}_N\} \subseteq \mathbb{R}^{N \times 3}$, normal vector set as $\mathbf{N} = \{\mathbf{n}_1, \mathbf{n}_2, \cdots, \mathbf{n}_N\} \subseteq \mathbb{R}^{N \times 3}$. Assume $\mathbf{x}_i$ as our interest point, and the objective is to extract geometric features for point $\mathbf{x}_i$. Then, the local reference frame $\{\mathbf{u}, \mathbf{v}, \mathbf{w}\}$ [37] for $\{(\mathbf{x}_i, \mathbf{x}_j), i \neq j\}$ is defined as:

$$
\begin{cases}
\mathbf{u} = \mathbf{n}_i \\
\mathbf{v} = \dfrac{(\mathbf{x}_j - \mathbf{x}_i) \times \mathbf{u}}{||(\mathbf{x}_j - \mathbf{x}_i) \times \mathbf{u}||} \\
\mathbf{w} = \dfrac{\mathbf{u} \times \mathbf{v}}{||\mathbf{u} \times \mathbf{v}||}
\end{cases}
\quad . \tag{1}
$$

Although Equ. 1 achieves the construction of local reference frames, it comes with two major problems. Firstly, it relies on normal vectors, which are often unavailable in many benchmarks [7, 48] and real-life scenarios. Despite normal estimation [13] is feasible, its computational cost escalates significantly with dense point clouds, and its accuracy diminishes considerably with sparse point clouds. Secondly, it involves multiple cross-product operations sequentially, which cannot be effectively parallelized in terms of tensor operations. This leads to the inevitable computational overheads.

To circumvent normal estimation and overcome the computational bottlenecks, we design approximated local reference frames through *Local Reference Constructor* (*LRCon*) module. Within each cluster generated by *Hierarchical Sampling* module, we establish point pairs between centroid and neighboring points. For each point pair, *LRCon* module leverages two adjacent neighbors along with their cross-product to serve as the approximate local reference frames. Denote number of neighbors as $K$, neighbors of centroid $\mathbf{x}_i$ as $\mathbf{X}_i = \{\mathbf{x}_{i1}, \mathbf{x}_{i2}, \cdots, \mathbf{x}_{iK}\} \subseteq \mathbb{R}^{K \times 3}$. Based on this setting, we can construct the approximated local reference frame $\{\hat{\mathbf{u}}, \hat{\mathbf{v}}, \hat{\mathbf{w}}\}$ for point pair $\{(\mathbf{x}_i, \mathbf{x}_{ij}), j = 1, 2, \cdots, K\}$, which is defined as:

$$
\begin{cases}
\hat{\mathbf{u}} = \dfrac{\mathbf{x}_{ij}^+ - \mathbf{x}_i}{||\mathbf{x}_{ij}^+ - \mathbf{x}_i||} \\
\hat{\mathbf{v}} = \dfrac{\mathbf{x}_{ij} - \mathbf{x}_i}{||\mathbf{x}_{ij}^- - \mathbf{x}_i||} \\
\hat{\mathbf{w}} = \dfrac{\hat{\mathbf{u}} \times \hat{\mathbf{v}}}{||\hat{\mathbf{u}} \times \hat{\mathbf{v}}||}
\end{cases}
\quad , \tag{2}
$$

where $\mathbf{x}_{ij}^+, \mathbf{x}_{ij}^-$ are the most adjacent points for $\mathbf{x}_{ij}$ in neighbor set $\mathbf{X}_i$ clockwise and counterclockwise. To maintain the consistency of local frame orientation, we apply clockwise cross-product [33] to compute $\hat{\mathbf{w}}$. When setting up the approximated local reference frames, the *LRCon* module basically finds the adjacent neighbors from the corresponding triangles in the triangle sets, *i.e.*, $\Delta_1 = \{\mathbf{x}_i, \mathbf{x}_{ij}, \mathbf{x}_{ij}^+\}$ and $\Delta_2 = \{\mathbf{x}_i, \mathbf{x}_{ij}, \mathbf{x}_{ij}^-\}$. As mentioned earlier, the *Hierarchical Sampling* module can boost the quality of triangle sets, ensuring the robustness against noisy points. This implicitly guarantees the reliability of the approximated local reference frames.

This approximation scheme allows us to establish local reference frames that are independent with normal vectors of point clouds. By re-ordering these neighbors in $\mathbf{X}_i$ based on their projected angles in $xy$-plane, we can efficiently derive approximated reference frames through tensor operations. Notably, with the integration of the

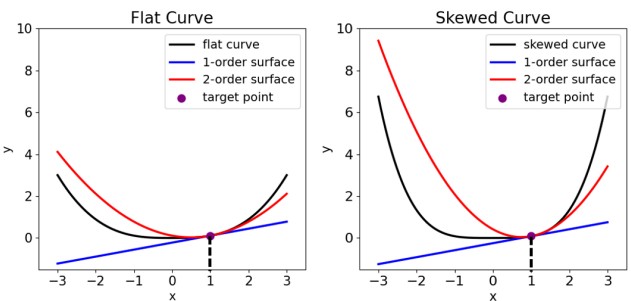

**Figure 3: Local shape comparison between a flat (left) and a skewed (right) 2D curve around point $x = 1$. Black represents the original curve, blue represents 1-order Taylor approximation, and red represents 2-order Taylor approximation.**

*LRCon* module, our OPFR only requires an additional **1.56ms** for inference, making it **65×** faster than vanilla PFH. Furthermore, since *LRCon* module eliminates the need of normal estimation, it is compatible with point clouds of varying densities.

### 3.3 Curve Feature Generator

We propose to approximate the local curve at point $(t, f(t))$ by excluding high-order derivatives using Taylor Series [43]:

$$
f(x) \approx \underbrace{f(t)}_{\text{location}} + \underbrace{f'(t)}_{\text{orientation}} (x - t) + \frac{1}{2} \underbrace{f''(t)}_{\text{curvature}} (x - t)^2. \tag{3}
$$

Intuitively, the derivatives $f'(t)$ and $f''(t)$ can reflect how the local curve is oriented and skewed near point $(t, f(t))$ respectively. From this Taylor approximation, it can be observed that, 1-order derivative information is inadequate for accurately characterizing local curves. To illustrate this point, consider the 2D example shown in Fig. 3. In the left figure, the curve may appear relatively flat compared to the right figure. However, both curves share the same normal vector (1-order derivative) at point $x = 1$, implying that these two points possess identical orientation. Therefore, if we neglect 2-order derivative, it would lead to confusion between these two distinct curves. In fact, these curves exhibit significant differences in how surfaces curve at point $x = 1$.

To exploit 2-order curvature information, we propose the *Curve Feature Generator* (*CFGen*) module. This module processes input point pairs along with their approximated local reference frames, generating geometric features that encompass location, orientation, and curvature. Denote the approximate local reference frames for $(\mathbf{x}_i, \mathbf{x}_{ij})$ as $\{\hat{\mathbf{u}}_{ij}, \hat{\mathbf{v}}_{ij}, \hat{\mathbf{w}}_{ij}\}$. The location and orientation can be naturally [33] characterized by relative position $\mathbf{x}'_{ij} = \mathbf{x}_{ij} - \mathbf{x}_i$ and frame cross-product $\mathbf{n}_{ij} = \hat{\mathbf{u}}_{ij} \times \hat{\mathbf{v}}_{ij}$, respectively. Furthermore, we propose the curvature proxy $\mathbf{p}_{ij}$ for point clouds, which is an approximation of curvature definition [40] from differential geometry. We provide the theoretical analysis for this part in the supplementary material. The curvature proxy $\mathbf{p}_{ij}$ is defined as:

$$
\mathbf{p}_{ij} = \frac{1}{||\mathbf{x}'_{ij}||} \cdot \arccos([\hat{\mathbf{u}}_{ij}; \hat{\mathbf{v}}_{ij}; \hat{\mathbf{w}}_{ij}] \odot \frac{\mathbf{x}'_{ij}}{||\mathbf{x}'_{ij}||}), \tag{4}
$$

Table 1: Performance of classification on ModelNet40 and ScanObjectNN. We evaluate different approaches in terms of overall accuracy (OA, %), mean per-class accuracy (mAcc, %), number of parameters (#Params) and FLOPs. Bold means outperforming other models on corresponding dataset. Green means an improvement from our OPFR compared with the original backbone.

| Method | Input | ModelNet40 | | ScanObjectNN | | #Params | FLOPs$^{\dagger}$ |
| --- | --- | --- | --- | --- | --- | --- | --- |
| | | OA | mAcc | OA | mAcc | | |
| PointNet [30] | 1k pnts | 89.2 | 86.0 | 68.2 | 63.4 | 3.47M | 0.45G |
| DGCNN [50] | 1k pnts | 92.9 | 90.2 | 78.1 | 73.6 | 1.82M | 2.43G |
| KPConv [47] | ~7k pnts | 92.9 | - | - | - | 14.3M | - |
| MVTN [9] | multi-view | 93.8 | **92.0** | 82.8 | - | 4.24M | 1.78G |
| RPNet [34] | 1k pnts* | 94.1 | - | - | - | 2.70M | 3.90G |
| CurveNet [57] | 1k pnts | 94.2 | - | - | - | 2.14M | 0.66G |
| RepSurf [33] | 1k pnts | 94.4 | 91.4 | 84.3 | 81.3 | 1.483M | 1.77G |
| RepSurf$^{\circ}$ [33] | 1k pnts | - | - | 86.0 | 83.1 | 6.806M | 4.84G |
| PointMLP [24] | 1k pnts | 94.1 | 91.5 | 85.4 | 83.9 | 12.6M | 31.4G |
| PointTrans. V2 [55] | 1k pnts* | 94.2 | 91.6 | - | - | - | - |
| PointNeXt [32] | 1k pnts | 93.2 | 90.8 | 87.7 | 85.8 | 4.5M | 6.5G |
| SPoTr [29] | 1k pnts | 93.2 | 90.8 | **88.6** | **86.8** | 3.3M | 12.3G |
| PointNet++ [31] | 1k pnts | 90.7 | 88.4 | 77.9 | 75.4 | 1.475M | 1.7G |
| **PointNet++ & OPFR (ours)** | 1k pnts | 94.5 ↑3.8 | 91.6 ↑3.2 | 85.7 ↑7.8 | 83.8 ↑8.4 | 1.487M | 1.85G |
| **PointNet++ & OPFR$^{\circ}$ (ours)** | 1k pnts | 94.6 ↑3.9 | 91.8 ↑3.4 | 88.5 ↑10.6 | 86.6 ↑11.2 | 8.42M | 5.9G |
| PointTrans. [65] | 1k pnts | 93.7 | 90.6 | 82.3 | 80.7 | 5.187M | 0.29G |
| **PointTrans. & OPFR (ours)** | 1k pnts | **94.8** ↑1.1 | **92.0** ↑1.4 | 88.1 ↑5.8 | 86.3 ↑5.6 | 5.190M | 0.33G |

∗: w/ normal vector. ◦: w/ double channels and deeper networks. †: FLOPs from 1024 input point cloud points.

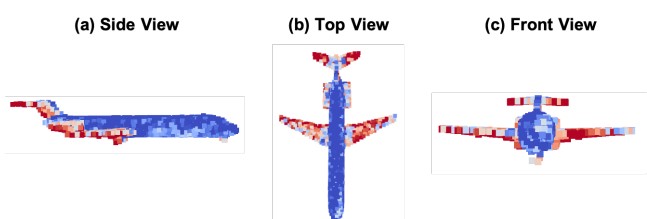

**(a) Side View**   **(b) Top View**   **(c) Front View**

**Figure 4: Three-view drawing of OPFR values for an airplane. We visualize the OPFR value for 1-st channel. Blue indicates small OPFR value and red indicates large OPFR value.**

where ⊙ is the entry-wise dot product. Note that, curvature proxy $\mathbf{p}_{ij}$ approximates the limit definition of curvature from differential geometry, making it inherently curvature-aware. Intuitively, $\mathbf{p}_{ij}$ effectively captures how the surface is curved in three reference frames $\{\hat{\mathbf{u}}_{ij}, \hat{\mathbf{v}}_{ij}, \hat{\mathbf{w}}_{ij}\}$ in terms of normalized angles.

### 3.4 On-the-fly Point Feature Representation

Point Feature Histogram (PFH) [37] utilizes histogram operations to aggregate regional geometric features and generate final representation for each point. We argue that, these predefined transformation functions are task-agnostic, which making the final representations not fitting well for specific tasks. To this end, motivated by PointNet++ [31], we employ shared-MLP to learn the final representations from point clouds. Therefore, the proposed OPFR representation $\mathbf{r}_i$ for point $\mathbf{x}_i$ is defined as:

$$\mathbf{r}_i = \mathcal{A}(\{\mathcal{F}([\mathbf{x}'_{ij}; \mathbf{n}_{ij}; \mathbf{p}_{ij}]) : j = 1, 2, \cdots, K\}), \quad (5)$$

where $\mathcal{A}$ is a pooling operation (*e.g.*, sum), $\mathcal{F}$ is a shared-MLP, and $[\mathbf{x}'_{ij}; \mathbf{n}_{ij}; \mathbf{p}_{ij}]$ are explicit geometric features obtained from *CFGen* module for one point pair $(\mathbf{x}_i, \mathbf{x}_{ij})$. By feeding OPFR representation $\mathbf{r}_i$ along with coordinate $\mathbf{x}_i$ into the backbone, the whole learning process can be achieved through end-to-end training.

In Fig. 4, the three-view drawing depicts the OPFR values of 1-st channel for an airplane. The blue hues represent areas with smaller OPFR values, typically the airplane body, while the red hues indicate larger OPFR values, primarily associated with the airplane wings. This color differentiation underlines our OPFR is sensitive to curvature variation across the airplane's structure, demonstrating the curvature-aware property of OPFR. We provide more visualization examples in the supplementary material. Additionally, it is important to highlight that, as shown in Fig. 5, our OPFR can outperform vanilla PFH by a large margin with the help of learnable shared-MLP. Furthermore, the introducing of shared-MLP only increases **0.012M** learnable parameters, which is approximately negligible for most popular backbones [31, 65].

### 4 EXPERIMENTS

We evaluate our OPFR on two primary tasks: point cloud classification and semantic segmentation. We choose two representative point cloud understanding models, PointNet++ [31] and Point Transformer [65], as our backbones to evaluate the effectiveness and compatibility of OPFR representations across different backbone architectures. Additionally, we carry out ablation studies to demonstrate the effectiveness of our OPFR network designs and quantitatively evaluate the efficiency and quality of OPFR feature representations. Moreover, due to space constraints, we present qualitative results in the supplementary material.

**Table 2: Performance of semantic segmentation on S3DIS 6-fold and S3DIS Area-5 benchmarks. We evaluate different approaches in terms of mean Intersection over Union (mIoU, %), mean per-class accuracy (mAcc, %), overall accuracy (OA, %), number of parameters (#Params) and FLOPs. Bold means outperforming other models on corresponding dataset. Green means an improvement from our OPFR compared with the original backbone.**

| Method | S3DIS 6-fold | | | S3DIS Area-5 | | | #Params | FLOPs[†] |
|---|---|---|---|---|---|---|---|---|
| | mIoU | mAcc | OA | mIoU | mAcc | OA | | |
| PointNet [30] | 47.6 | 66.2 | 78.5 | 41.1 | 48.9 | - | 1.7M | 4.1G |
| KPConv [47] | 70.6 | 79.1 | - | 67.1 | 72.8 | - | 14.9M | - |
| RPNet [34] | 70.8 | - | - | - | - | - | 2.4M | 5.1G |
| RepSurf [33] | 74.3 | 82.6 | 90.8 | 68.9 | 76.0 | 90.2 | 0.976M | 6.7G |
| PointTrans. V2 [55] | - | - | - | 71.6 | 77.9 | 91.1 | - | - |
| PointNeXt-B [32] | 71.5 | - | 88.8 | 67.3 | - | 89.4 | 3.8M | 8.9G |
| PointNeXt-XL [32] | 74.9 | - | 90.3 | 70.5 | - | 90.6 | 41.6M | 84.8G |
| Superpoint Trans. [41] | 76.0 | 85.5 | 90.4 | 68.9 | 77.3 | 89.5 | 0.21M | - |
| ConDaFormer* [3] | - | - | - | **72.6** | 78.4 | 91.6 | - | - |
| PointNet++ [31] | 59.9 | 66.1 | 87.5 | 56.0 | 61.2 | 86.4 | 0.969M | 7.2G |
| **PointNet++ & OPFR (ours)** | 74.6 ↑14.7 | 83.0 ↑16.9 | 90.5 ↑3.0 | 69.1 ↑13.1 | 76.9 ↑15.7 | 90.0 ↑3.6 | 0.979M | 7.5G |
| PointTrans. [65] | 73.5 | 81.9 | 90.2 | 70.4 | 76.5 | 90.8 | 7.768 M | 5.8G |
| **PointTrans. & OPFR (ours)** | **76.9** ↑3.4 | **85.6** ↑3.7 | **92.0** ↑1.8 | **72.6** ↑2.2 | **78.6** ↑2.1 | **91.7** ↑0.9 | 7.771M | 6.4G |

∗: w/o test-time-augmentation. †: FLOPs from 15000 input point cloud points.

**Table 3: Semantics segmentation results for each class on S3DIS Area-5. We evaluate model performance in terms of mean accuracy (mIoU, %) for each semantic class. Bold means top improved semantic classes in terms of mIoU. Green means an improvement from our OPFR compared with the original backbone.**

| Method | ceiling | floor | wall | beam | **column** | window | door | chair | table | bookcase | sofa | board | clutter | mIoU |
|---|---|---|---|---|---|---|---|---|---|---|---|---|---|---|
| PointNet++ [31] | 91.47 | 98.18 | 82.19 | 0.00 | **17.99** | 57.75 | 64.64 | 79.70 | 87.82 | 67.11 | 69.76 | 65.29 | 50.79 | 56.0 |
| **PointNet++ & OPFR (ours)** | 93.13 | 98.37 | 85.38 | 0.00 | **41.50** ↑23.51 | 62.32 | 71.56 | 80.37 | 89.86 | 77.25 | 72.67 | 68.18 | 57.12 | 69.1 |
| PointTrans [65] | 93.71 | 98.00 | 86.78 | 0.00 | **36.35** | 64.79 | 73.40 | 83.30 | 89.84 | 68.80 | 73.32 | 74.33 | 58.17 | 70.4 |
| **PointTrans. & OPFR (ours)** | 93.68 | 98.11 | 88.20 | 0.00 | **55.16** ↑18.81 | 69.02 | 73.53 | 83.68 | 90.43 | 75.57 | 79.71 | 75.67 | 62.06 | 72.6 |

**Implementation details.** For the *Hierarchical Sampling* module, we set $k_1 = 20$ and $k_2 = 4$ to control the number of candidate centroids and selected centroids respectively. The shared-MLP consists of three layers with 30 OPFR dimensions ($\mathbf{r}_i \in \mathbb{R}^{30}$), followed by a sum pooling operation. These are achieved via empirical studies, which will be further discussed in Sec. 4.3. Following RepSurf [33], we set $k_3 = 8$, considering the trade-off between performance and efficiency. We use CrossEntropy loss and label smoothing [44] techniques with a ratio of 0.3 for both tasks. We provide more details about implementation in the supplementary material.

## 4.1 Classification

We evaluate our OPFR on two commonly used benchmarks for point cloud classification: ModelNet40 [56] and ScanObjectNN [48].
**Experimental setups.** Following RepSurf [33], we implement two versions to integrate OPFR with PointNet++ [31], one standard version and one scaled-up version. The scaled-up version doubles the channels of standard version and exploits deeper networks. If not specified, we default to the standard version. We also apply the channel de-differentiation design [33] when integrated with PointNet++. We opt Adam [16] optimizer with default parameters to train our models for 250 epochs with a batch size of 64 and initial learning rate of 0.002. We apply exponential learning rate

decay scheme with decay rate of 0.7. The whole training and testing process are conducted through one NVIDIA Quadro P5000 16GB GPU. For evaluation metrics, we use overall accuracy (OA) and mean accuracy within each classes (mAcc). For efficiency metrics, we use number of learnable parameters (#Params) and floating point operations (FLOPs). For a fair comparison, we calculate FLOPs from 1024 input point clouds, and utilize single-scale grouping (SSG) set abstraction [31] for all PointNet++ based [24, 31–33] methods.
**Classification on ModelNet40.** ModelNet40 [56] is one synthetic object classification benchmark, which contains 9843 training samples and 2468 testing samples. They contain 100 unique CAD models from 40 object categories. The experimental results are presented in Tab. 1. The results reveal that our OPFR significantly improves PointNet++ [31] backbone by 3.8% OA and 3.2% mAcc, with just an additional 0.012M more parameters and 0.15G more FLOPs. The scaled-up OPFR further attains a slight improvement of 0.1% OA and 0.2% mAcc. Moreover, when integrated with transformer-based backbone, Point Transformer [65], our OPFR achieves the state-of-the-art 94.8% OA and 92.0% mAcc (+1.1% OA and +1.4% mAcc).
**Classification on ScanObjectNN.** ScanObjectNN [48] is a challenging, real-world object classification benchmark. It is composed of 2902 point cloud samples from 15 categories, including occlusion and background. Following the typical protocol [29, 31, 33], we

verify our OPFR on the hardest variant (PB_T50_RS_variant) of ScanObjectNN. In Tab. 1, the proposed OPFR achieves 85.7% OA and 83.8% mAcc (+7.8% OA and +8.4% mAcc) on PointNet++ backbone, which outperforms RepSurf [33] by a large margin of 1.4% OA and 2.5% mAcc with comparable model size. Our result surpasses PointMLP [24] by 0.3% OA as well, and utilizes 9× fewer parameters. Furthermore, we scale up our proposed OPFR and achieve 88.5% OA and 86.6% mAcc, which demonstrates a superiority of 0.8% OA and 0.8% mAcc compared with state-of-the-art MLP-based backbone, PointNeXt [32]. Our result is also comparable to prior state-of-the-art transformer-based backbone, SPoTr [29], with around 2.1× fewer FLOPs. When integrated with Point Transformer, our OPFR attains a notable improvement of 5.8% OA and 5.6% mAcc, which only increases 0.003M more parameters and 0.04G more FLOPs.

## 4.2 Semantic Segmentation

We evaluate our proposed OPFR representations on a challenging benchmark, S3DIS [1], for semantic segmentation task.

**Experimental setups.** When integrated with PointNet++ [31], we apply the channel de-differentiation design [33]. We opt AdamW [23] with default parameters to train our models for 100 epochs with a batch size of 8 and initial learning rate of 0.006. Here, we employ multi-step learning rate decay scheme and decay at [60,80] epochs with a decay rate of 0.1. The whole training and testing process are conducted through two NVIDIA A40 48GB GPU. For evaluation metrics, we use mean of classwise intersection over union (mIoU), mean of classwise accuracy (mAcc), and overall accuracy (OA). For a fair comparison, we calculate FLOPs from 15000 input point clouds [32], and leave test-time-augmentation [3] in absence.

**Semantic Segmentation on S3DIS.** S3DIS [1] encompasses 271 scenes which are distributed across 6 indoor areas, with each individual point being classified into one of 13 semantic labels. Following a common protocol [31, 45], we evaluate the presented approach in two modes: (a) Area-5 is withheld for training and is used for testing, and (b) 6-fold cross-validation. In Tab. 2, our proposed OPFR considerably enhances PointNet++ [31] by 14.7%/16.9%/3.0% (mIoU/mAcc/OA) on S3DIS 6-fold benchmark. Our result is comparable to PointNeXt-XL [32], with around 40× fewer parameters and 11× fewer FLOPs. When integrated with Point Transformer [65], the performance of OPFR exceeds previous state-of-the-art Superpoint Transformer [41] by 0.9%/0.1%/1.6% (mIoU/mAcc/OA) for S3DIS 6-fold. Meanwhile, on S3DIS Area-5, our OPFR attains mIoU/mAcc/OA of 72.6%/78.6%/91.7% (+2.2%/+2.1%/+0.9%), surpassing the prior state-of-the-art ConDaFormer [3].

Furthermore, as shown in Tab. 3, we present quantitative segmentation results for each semantic class on S3DIS Area-5 in terms of mIoU. In Tab. 3, the top performance gain comes from the most challenging **columns** semantic class for both PointNet++ and Point Transformer backbones. Within all classes **columns** exhibit a distinct columnar structure, which consists of two or three planes in S3DIS dataset. This multi-plane structure can be effectively captured by different clusters generated from the proposed *Hierarchical Sampling* module, which facilitates the recognition of **column** pattern with greater ease. Furthermore, we provide qualitative results in the supplementary material.

**Table 4: Ablation study on the effectiveness of different modules. We conduct experiments on ScanObjectNN dataset.**

| Method | OA | mAcc |
|---|---|---|
| **PointNet++ & OPFR (ours)** | **85.68** | **83.81** |
| (−) *Hierarchical Sampling* strategy | -1.17 | -0.91 |
| (−) *Curve Feature Generator* | -2.02 | -1.76 |
| (−) shared-MLP | -1.53 | -1.34 |

**Table 5: Ablation study on the designs of OPFR netowrk architecture. We conduct experiments on ScanObjectNN dataset. (#(OPFR dims): number of OPFR dimensions, #(layers): number of shared-MLP layers)**

| Pooling | BN | #(OPFR dims) | #(layers) | OA |
|---|---|---|---|---|
| max | ✓ | 30 | 3 | 85.47 |
| avg | ✓ | 30 | 3 | 85.55 |
| sum | ✓ | 30 | 3 | **85.68** |
| sum | ✗ | 30 | 3 | 85.32 |
| sum | ✓ | 30 | 3 | **85.68** |
| sum | ✓ | 10 | 3 | 85.32 |
| sum | ✓ | 30 | 3 | **85.68** |
| sum | ✓ | 64 | 3 | 85.44 |
| sum | ✓ | 128 | 3 | 84.54 |
| sum | ✓ | 30 | 1 | 83.34 |
| sum | ✓ | 30 | 2 | 84.89 |
| sum | ✓ | 30 | 3 | **85.68** |
| sum | ✓ | 30 | 4 | 85.42 |
| sum | ✓ | 30 | 5 | 84.50 |

## 4.3 Ablation Study

We ablate some critical designs of our standard OPFR with PointNet++ [31] backbone on ModelNet40 [56] and ScanObjectNN [48] dataset for an insightful exploration.

**Effectiveness of different OPFR modules.** Shown in Tab. 4, as we remove *Hierarchical Sampling* module, *Curve Feature Generator* module, and 3-layer shared-MLP, the overall accuracy (OA) decreases by 1.17%, 2.02%, 1.53% and mean accuracy (mAcc) drops by 0.91%, 1.76%, 1.34% respectively. From this empirical study, we can confirm that, explicit geometric features are crucial for 3D object understanding, and shared-MLP is necessary as well to enhance the semantics of obtained geometric features. Furthermore, due to the use of *Hierarchical Sampling* module, we can effectively release the distortion of triangle sets, thereby improving the quality of geometric features. Additionally, we argue that, the *Hierarchical Sampling* module can be applied to RepSurf [33] to handle the distorted triangle sets from $k$ nearest neighbors. Due to space limits, we provide the ablation study in the supplementary material.

**Designs of OPFR network architecture.** We ablate the designs of OPFR network architecture in terms of pooling operation $\mathcal{A}$ and shared-MLP $\mathcal{F}$ in Tab. 5. Empirical results demonstrate that, usage of summation pooling, batch normalization, and three-layer shared-MLP with 30 OPFR dimensions outperforms other options. From our experiments, we hypothesize that, the network tends to

**Table 6: Ablation study on the hyper-parameters sensitivity. We evaluate the overall accuracy (OA) on ScanObjectNN for different combinations of hyper-parameters $k_1$ and $k_2$.**

| OA | $k_1 = 10$ | $k_1 = 20$ | $k_1 = 40$ | $k_1 = 60$ |
|---|---|---|---|---|
| $k_2 = 2$ | 85.31 | 85.52 | 85.41 | 85.32 |
| $k_2 = 4$ | 85.43 | **85.68** | 85.55 | 85.42 |
| $k_2 = 6$ | 85.51 | 85.66 | 85.51 | 85.40 |
| $k_2 = 8$ | 85.47 | 85.61 | 85.52 | 85.46 |

**Table 7: Ablation study on the efficiency of OPFR representations. We test the speed of all methods with one NVIDIA A40 GPU. (#(Extra Params): number of extra parameters, Infer Speed: inference duration per input sample)**

| Method | #(Extra Params) | Infer Speed |
|---|---|---|
| PointNet++ & PFH [35] | - | 102ms |
| PointNet++ & RepSurf [33] | 0.008M | 1.12ms |
| **PointNet++ & OPFR (ours)** | **0.012M** | **1.56ms** |

encounter overfitting issues as we increase the number of OPFR dimensions and shared-MLP layers.

**Sensitivity of hyper-parameters.** In *Hierarchical Sampling* module, we are required to determine number of surface centroid candidates $k_1$, number of selected surface centroids $k_2$ and number of neighbors $k_3$. Following RepSurf [33] design, we fix $k_3$ equal to 8 to construct OPFR and explore the relation between $k_1$ and $k_2$ in terms of overall accuracy (OA) in Tab. 6. Generally speaking, our OPFR is relatively insensitive to the choices of hyper-parameters. As the value of $k_1$ increases, there is an initial rise in overall accuracy, which is subsequently followed by a slight decline. We hypothesize that, this phenomenon is attributed to the inherent trade-off between exploration and concentration. When $k_1$ is small, we are unable to capture the local region of point clouds effectively. Conversely, when $k_1$ is too large, we move far from the original point, leading to the deviation of obtained geometric features. Furthermore, our OPFR is insensitive to the change of $k_2$. We hypothesize that, this behavior primarily stems from these $k_2$ clusters may overlap with each other. To avoid computation overheads, we consider $(k_1 = 20, k_2 = 4)$ as an ideal choice.

**Efficiency of OPFR representations.** Shown in Tab. 7, we evaluate the efficiency of our OPFR representations in terms of number of extra parameters and inference speed. Empirically, although vanilla PFH introduces no extra learnable parameters, it requires 102ms for each input sample to generate the final representation, rendering it impractical for online network training. The main computational bottlenecks lie in the estimation of point clouds normal vectors [13, 52]. We propose novel *Local Reference Constructor* module to eliminate the needs of normal estimation and overcome the computational overheads. We achieve an impressive inference speed of **1.56ms (65× faster)** with a marginal increase of **0.012M** number of parameters. Therefore, OPFR can serve as a versatile plug-and-play module for various backbones. Furthermore, the efficiency of our OPFR is close to the previous state-of-the-art plug-and-play feature

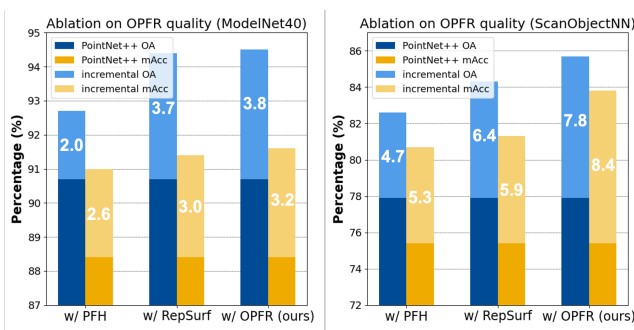

**Figure 5: Ablation study on the quality of OPFR representations. We evaluate the feature quality in terms of overall accuracy (OA, %) and mean accuracy (mAcc, %). White means an improvement from PointNet++ backbone.**

representation RepSurf [33], with only 0.004M more parameters and 0.44ms more inference time.

**Quality of OPFR representations.** Shown in Fig. 5, we compare the performance between PFH [37], RepSurf [33], and proposed OPFR using PointNet++ [31] backbone. All of them are injected to PointNet++ as extra features. By incorporating vanilla PFH, overall accuracy (OA) and mean accuracy (mAcc) are enhanced by 2.0% and 2.6% on ModelNet40, 4.7% and 5.3% on ScanObjectNN, emphasizing the effectiveness of regional curvature knowledge. This gain further escalates to 3.8% and 3.2% on ModelNet40, 7.8% and 8.4% on ScanObjectNN in OA and mAcc respectively, when equipped with the proposed OPFR. This demonstrates the significance of shared-MLP, which enriches the obtained geometric features. Furthermore, compared with the previous state-of-the-art feature representation RepSurf, our OPFR outperforms it dramatically on ScanObjectNN, with a considerable margin of 1.4% and 2.5% higher OA and mAcc. We hypothesize that, this phenomenon is attributed to the uses of explicit curvature knowledge and robust sampling strategy, which are underexplored in RepSurf.

## 5 CONCLUSION

We propose the novel plug-and-play module ***On-the-fly Point Feature Representation (OPFR)*** for various backbones. It explicitly captures local geometry including location, orientation and curvature through *Curve Feature Generator* module. We further develop the *Local Reference Constructor* module to improve efficiency and enable on-the-fly processing. Additionally, we introduce the *Hierarchical Sampling* module to mitigate the distortion of triangle sets that occurs in the naive $k$ nearest neighbors sampling, thereby enhancing the robustness of obtained geometric features. We evaluate the proposed OPFR on ModelNet40 [56] and ScanObjectNN [48] benchmarks for point cloud classification task, S3DIS [1] for semantic segmentation task. For both PointNet++ [31] and Point Transformer [65] backbones, our presented OPFR achieves the state-of-the-art results on different benchmarks. The comprehensive empirical results demonstrate the backbone-agnostic nature of our proposed method. We believe that our work can prompt consideration of how to better leverage geometric knowledge in network architecture designs for understanding point clouds.

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
