# OpenReview forum: "On-the-fly Point Feature Representation for Point Clouds Analysis"
_acmmm.org/ACMMM/2024/Conference — MM2024 Poster_

### Official Review · Reviewer_gZRz · 2024-05-24

**Rating:** 5
**Confidence:** 2

**Summary:**

This paper addresses the challenges in point cloud analysis due to their unorderness, sparsity and irregularity. Traditional methods struggle with capturing explicit local geometry like curvature and orientation. The authors propose an On-the-fly Point Feature Representation (OPFR) to explicitly capture geometric information through a Curve Feature Generator module, inspired by Point Feature Histogram (PFH). The OPFR is efficient, requiring only 1.56ms for inference, making it 65 times faster than vanilla PFH. The proposed method significantly improves classification and semantic segmentation accuracy on various datasets.

**Strengths:**

1. **Efficiency**: The On-the-fly Point Feature Representation (OPFR) significantly improves computational efficiency, requiring only 1.56ms for inference, which is 65 times faster than the traditional Point Feature Histogram (PFH). This efficiency is crucial for real-time applications and large datasets.

2. **Explicit Geometric Information**: The OPFR captures explicit geometric information, including location, orientation, and curvature, which are essential for accurate point cloud analysis. This approach overcomes the limitations of previous methods that implicitly capture geometric information.

3. **Versatility and Compatibility**: The OPFR is designed as a plug-and-play module that can be integrated with various backbones, including MLP-based and Transformer-based architectures. This versatility allows it to enhance different types of neural network architectures used in point cloud analysis.

**Limitations:**

1. **Complexity of Hierarchical Sampling**: The Hierarchical Sampling module, while improving the quality of geometric features, introduces additional complexity in the preprocessing stage. This might limit its scalability for extremely large datasets or real-time applications where rapid data processing is crucial.
2. **Dependency on Accurate Triangle Sets**: The effectiveness of the Local Reference Constructor and Curve Feature Generator modules relies heavily on the accuracy and quality of the triangle sets generated.
3. **Sensitivity to Hyperparameters k1 and k2**: Experimental results show that OPFR is relatively insensitive to the choices of k1 and k2. Please provide a clearer explanation.

**Suitability:**

2

---

### Official Review · Reviewer_bv4F · 2024-05-25

**Rating:** 3
**Confidence:** 3

**Summary:**

In this paper, the authors introduce a novel module, On-the-fly Point Feature Representation (OPFR), which is designed to enhance point cloud analysis by explicitly capturing geometric information such as curvature and orientation. The OPFR module incorporates a Curve Feature Generator that builds upon the concept of the Point Feature Histogram (PFH), yet it is significantly optimized to improve performance. By introducing the Local Reference Constructor module, the authors manage to approximate local coordinate systems using triangle sets, enabling the OPFR module to operate much faster (65 times faster than the traditional PFH) and with minimal additional parameters.

**Strengths:**

1.	This paper introduces a novel Curve Feature Generator module that effectively leverages explicit features from point clouds, significantly improving performance.
2.	The methodology is presented clearly and in detail, making it easy to follow.
3.	The experiments conducted are both thorough and comprehensive.

**Limitations:**

1.	This paper experimentally follows the methodologies of RepSurf but does not include a comparison and analysis with RepSurf-U. Notably, some ideas presented in this paper bear similarities to those in RepSurf.
In Table 3, the values in the beam column are listed as 0. Is this indicative of actual zero results, or were these values possibly not filled in?

**Suitability:**

2

---

### Official Review · Reviewer_KAX7 · 2024-05-25

**Rating:** 4
**Confidence:** 2

**Summary:**

This paper presents a novel approach to point cloud representation and analysis through the introduction of the On-the-fly Point Feature Representation (OPFR) module. This module aims to provide a more explicit depiction of local geometric information within point clouds. The method is composed of three key components: Hierarchical Sampling, Local Reference Constructor, and Curve Feature Generator. Hierarchical Sampling improves triangle set generation robustness, while the Local Reference Constructor offers an alternative method for generating point clouds independent of local geometric data. The Curve Feature Generator utilizes second-order derivative curvature information to generate detailed curvature features. The paper demonstrates the effectiveness of OPFR across various benchmarks for point cloud classification and semantic segmentation tasks, achieving state-of-the-art results with different backbone architectures like PointNet++ and Point Transformer. Overall, the paper highlights the importance of integrating geometric knowledge into network architecture designs for better understanding and processing of point clouds.

**Strengths:**

The manuscript presents innovative approaches in leveraging curvature information and refining sampling algorithms. These methodologies are rigorously validated through a comprehensive array of experiments. The clarity and coherence of the presentation are commendable. It is noteworthy that the method exhibits strong reproducibility and simplicity, with no undue complexity.

**Limitations:**

The current state of the art remains uncertain, particularly with regard to the selection of the 2021 experimental baseline. Additionally, the paper referenced for the utilization of curvature information dates back to 2016, lacks comparison with more recent studies, and fails to convincingly demonstrate the advancement and effectiveness of curvature information utilization.

**Suitability:**

3

---

### Official Review · Reviewer_zgu5 · 2024-05-28

**Rating:** 4
**Confidence:** 3

**Summary:**

This paper proposes a local point feature representation, On-the-fly Point Feature Representation (OPFR), for point cloud analysis. The method consists of three main modules: Hierarchical Sampling, Local Reference Constructor and Curve Feature Generator. OPFR obtains the capability to explicitly capture not only location and orientation knowledge, but also curvature geometry. The experiments show that the method facilitates the classification and semantic segmentation performances.

**Strengths:**

- The local reference constructor utilizes the two nearest points (clockwise and counterclockwise) of a point pair to construct a local reference frame. It avoids using the normal vector to build the frame and improve the efficiency.

- The curve feature generator uses a curvature proxy to depict the surface curvature and add more geometric details to point features.

- The experiments show that OPFR can facilitate classification and semantic segmentation tasks based on different backbones.

**Limitations:**

- The Hierarchical Sampling is kind of trivial. The sampling procedure can directly use FPS to sample more points and create clusters by k-NN. The Hierarchical Sampling module doubles the sampling and clustering step, which is not necessary and more complicated.

- The efficiency of OPFR should be further tested.  The construction of a local reference frame is applied to each point pair in each local cluster, which is a little fussy. I am curious about the inference time change before and after adding the OPFR to PointNet++ and PointTransformer.

- The procedure for finding the clockwise and counterclockwise most adjacent points x_{ij}^+, x_{ij}^- needs more details. Besides, the efficiency of getting these points for each point pair needs further analysis.

**Suitability:**

2

---

### Meta-Review · Area_Chair_NwFX · 2024-06-28

**Recommendation:** Accept (Poster)
**Confidence:** 4

**Metareview:**

Initially the paper receives 3 positive but 1 negative ratings. After the rebuttal, all reviewers are satisfied with the authors' follow-up answers and efforts on addressing their questions. Particularly, as one reviewers raised the rating to borderline accept, all four reviewers reached a concensus on accpeting this submission. After carefully reading both the reviews and the paper, the AC agrees with the reviewers' recommendations, given its merits in presenting a new point cloud feature representation method with strong performances in fundamental 3D analysis tasks. The AC decides to recommend an acceptance, and suggests the authors to prepare a better camera ready paper by referring to some rebuttal content.

---

### Meta-Review · Senior_Area_Chairs · 2024-07-10

**Recommendation:** Accept (Poster)
**Confidence:** 4

**Metareview:**

This paper received mixed ratings initially. After rebuttal, all the reviewers tend to accept the paper. SAC and AC agree with reviewers and recommend acceptance of the paper.